# LEARNING DEEP IMPROVEMENT REPRESENTATION TO ACCELERATE EVOLUTIONARY OPTIMIZATION

## ABSTRACT

Evolutionary algorithms excel at versatile optimization for complex (e.g., multiobjective) problems but can be computationally expensive, especially in high-dimensional scenarios, and their stochastic nature of search may hinder swift convergence to global optima in promising directions. In this study, we train a multilayer perceptron (MLP) to learn the improvement representation of transitioning from poor-performing to better-performing solutions during evolutionary search, facilitating the rapid convergence of the evolutionary population towards global optimality along more promising paths. Then, through the iterative stacking of the well-trained lightweight MLP, a larger model can be constructed, enabling it to acquire deep improvement representations (DIR) of solutions. Conducting evolutionary search within the acquired DIR space significantly accelerates the population's convergence speed. Finally, the efficacy of DIR-guided search is validated by applying it to the two prevailing evolutionary operators, i.e., simulated binary crossover and differential evolution. The experimental findings demonstrate its capability to achieve rapid convergence in solving challenging large-scale multi-objective optimization problems.

## 1 INTRODUCTION

Optimization serves as a fundamental component in numerous real-world applications and machine learning algorithms. For instance, it plays an essential role in optimizing vehicle routes for cost-efficiency in logistics (Thanh et al., 2023), forms the core of hyperparameter tuning in AutoML (Zhang et al., 2023), defines and minimizes the multiple loss functions in multitask learning (Lin et al., 2019), etc. The optimization problems in these applications may be challenging due to their non-convex, multiobjective, evaluation-expensive, and/or large-scale nature. Addressing such challenges demands the use of well-designed optimizers, with evolutionary algorithms (EAs) standing out as promising problem-solving tools (Liu, 2022). Nevertheless, EAs can be computationally demanding, which limits their adaptability to lightweight optimization requirements (Coello Coello et al., 2020). In recent years, there has been a growing emphasis on conducting computations closer to data sources, such as onboard or alongside a connected camera in a self-driving car, to enable real-time optimization services (Gulotta, 2023). This shift has led to a transition of computing from the centralized cloud to the edge devices, where computing resources are severely limited.

However, many existing EAs were developed without considering these resource limitations. In the quest for lightweight optimization, EAs must enhance efficiency to address the growing complexity of challenges (Del Ser et al., 2019), notably those related to large model and big data optimization that are often computationally demanding, particularly in terms of function evaluations (Chugh et al., 2019). Building on the observations outlined above, this study aims to enhance the efficiency of EAs for solving large-scale multi-objective optimization problems (LMOPs). In the literature, extensive efforts have been dedicated to improve EAs for solving LMOPs, which can be broadly classified into three main categories:

**Decomposition of Search Space:** This approach employs a divide-and-conquer mechanism, where decision variables are grouped or clustered by the developed variable decomposition methods (Zhao et al., 2022), including linear, random, and differential based methods (Ou et al., 2022). Optimization is then carried out collaboratively on each of these groups (subspaces), simplifying the problem-solving process (Zhong et al., 2022). However, it typically relies on rich domain exper-

tise for problem decomposition which may not be available. Incorrect grouping of variables may mislead evolutionary search and slow down population convergence (Duan et al., 2023). Analyzing the importance (or contribution) of variables and their interrelationships before grouping requires a substantial number of function evaluations (Liu et al., 2022).

**Dimension Reduction of Search Space:** This method transforms the original LMOP into smaller-scale problems using existing dimensionality reduction technique, such as random embedding (Qian & Yu, 2017), unsupervised neural networks (Tian et al., 2020), problem transformation (Zille et al., 2016), and principal component analysis (Liu et al., 2020). This conversion allows optimization to take place in a simplified representation space, leading to a substantial reduction in the volume of the high-dimensional search space. Nevertheless, it does not guarantee the preservation of the original global or near-global optimum when operating within the compressed search space, and thus it may potentially miss certain optimal regions, making populations susceptible to local optima entrapment. The dimensionality reduction process often overlooks constraints related to computational resources.

**Design of Novel Search Strategy:** In contrast to the preceding methods that alleviate problem complexity before optimization, this category of algorithms tackles LMOPs directly, taking all decision variables into account. It achieves this by designing new, powerful evolutionary search strategies for offspring reproduction, such as competitive learning-based search (Liu et al., 2021), bidirectional-guided adaptive search (He et al., 2020a), adversarial learning-aided search (Wang et al., 2021b), and fuzzy-controlled search (Yang et al., 2021). Without proper guidance towards the correct search direction, there's a likelihood of venturing into the misleading areas during optimization, resulting in a wasteful consumption of computing resources (Omidvar et al., 2021). These novel search strategies still fall considerably short of meeting the demands for lightweight optimization.

Despite these efforts, their search capabilities often fall short of effectively handling the exponentially expanded search space within the constraints of acceptable computational resources. In pursuit of accelerated evolutionary optimization, researchers have investigated online innovization progress operators aimed at guiding offspring towards learned promising directions (Deb & Srinivasan, 2006). These operators involve training machine learning models online to get performance improvement representations of solutions (Gaur & Deb, 2017). This process encompasses three primary steps: gathering solutions from previous generations, training the model to identify patterns, and utilizing it to rectify newly generated offspring (Mittal et al., 2020). However, existing innovization operators are only developed for small-scale optimization. In addition, the online training of deep models introduces computational overhead, particularly in the context of large-scale optimization, and the resulting acceleration in convergence still falls short of expectations. In response, to expedite the optimization of LMOPs, this work introduces a deep accelerated evolutionary search strategy driven by an inexpensive large model, which is stacked repeatedly by multiple lightweight models. This study presents three main contributions: 1) Development of a lightweight model capable of learning both compressed and performance improvement representations of solutions. 2) Analysis of the varying impacts of evolutionary search in the learned representation space. 3) Design of a large model for acquiring deep improvement representations (DIR) of solutions, aimed at enabling efficient optimization of LMOPs. The relevant background, technical details, and specific experimental design and verification are respectively elaborated in sections 2, 3, and 4 below.

## 2 PRELIMINARIES AND MOTIVATIONS

### 2.1 LARGE-SCALE MULTIOBJECTIVE OPTIMIZATION

We exclusively assess the performance of EAs on continuous LMOPs. These LMOPs involve multiple conflicting objectives defined over high-dimensional solution vectors with a considerable number of interrelated variables. For simplicity and generalization, an LMOP is defined as follows:

$$\text{Minimize } F(x) = (f_1(x), \ldots, f_m(x)), x \in \Omega \tag{1}$$

where $x = (x_1, x_2, \ldots, x_n)$ is a solution vector with $n$ variables from the search space, and $F(x)$ defines $m$ objective functions $f_1(x), \ldots, f_m(x)$, $m \geq 2$ and $n$ is a relatively large value (e.g., $n \geq 1000$). Due to the inherent conflicts among these objectives, finding a single optimal solution for LMOPs is often unattainable. Instead, LMOPs typically yield a set of trade-off solutions known as the Pareto set (PS). Moreover, the projection of this PS onto the objective space is termed the Pareto front (PF). Consequently, the primary goal when addressing an LMOP with EAs is to discover

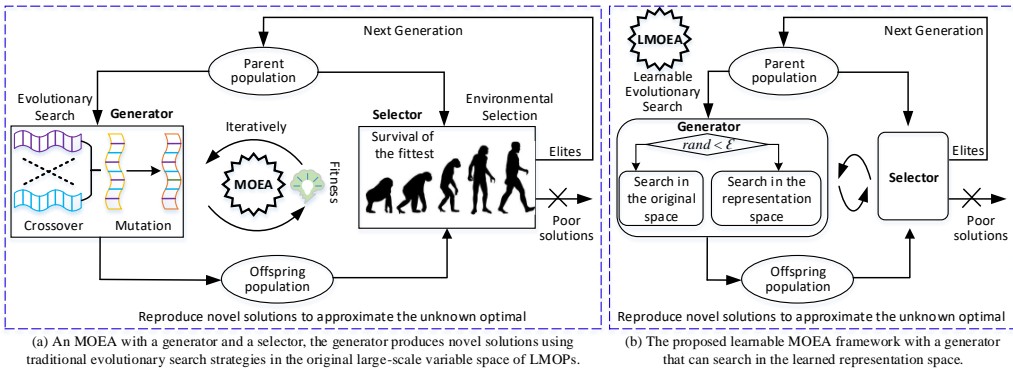

(a) An MOEA with a generator and a selector, the generator produces novel solutions using traditional evolutionary search strategies in the original large-scale variable space of LMOPs.

(b) The proposed learnable MOEA framework with a generator that can search in the learned representation space.

Figure 1: Illustration of the main process in an MOEA and the proposed Learnable MOEA.

a set of solutions that effectively and evenly approximate the PS/PF. To facilitate a comprehensive understanding of solving LMOPs, we introduce two key definitions:

**Definition 1 (Pareto Dominance):** given two solutions $x$ and $y$. we say $x$ dominates $y$, termed as $x \prec y$, if $f_i(x) \leq f_i(y)$ for $\forall i \in \{1, 2, \ldots, m\}$ and $f_j(x) < f_j(y)$ that for $\exists j \in \{1, 2, \ldots, m\}$.

**Definition 2 (Pareto Optimal Solution):** we say solution $x^*$ is a Pareto optimal if and only if $x^*$ cannot be dominated by any solution $x \in \Omega$.

## 2.2 MULTIOBJECTIVE EVOLUTIONARY ALGORITHMS

Multiobjective evolutionary algorithms (MOEAs) have gained widespread popularity in tackling complex multiobjective optimization problems (Guliashki et al., 2009). As shown in Figure 1(a), an MOEA begins with an initial parent population and generates novel offspring using a generative model equipped with evolutionary operators, such as crossover and mutation. These parent and offspring solutions are then evaluated by a selective model, which retains only the elite solutions identified as superior for survival into the next generation. Interestingly, this MOEA approach shares common traits with other problem-solving models like generative adversarial networks (Goodfellow et al., 2014) and reinforcement learning (Wang et al., 2021a). Specifically, an MOEA's generator aims to produce offspring with higher quality than their parents, while its selector classifies solutions based on their quality, subsequently filtering out poorly performing ones. Together, the generator and selector constitute a synergistic mechanism driving the search for diverse and increasingly convergent solutions to approximate elusive optima.

Despite significant development over the years, MOEAs still face limitations in effectively addressing LMOPs. The challenges can be attributed to several factors. As the number of variables increases, the search space grows exponentially, demanding that the generator exhibit enhanced search capabilities, such as accelerated convergence, while working within limited computational resources. Moreover, the intricate structural and property characteristics of LMOPs, including factors like separability and nonlinearity, complicate matters further. Consequently, effective search strategies employed by the generator must be scalable to combat the "curse of dimensionality" inherent in extensive search spaces (Liu, 2022). Unfortunately, conventional evolutionary operators like simulated binary crossover (SBX), polynomial mutation (PM), particle swarm optimization, differential evolution (DE), and evolutionary strategy have been proven ineffective when confronted with the challenges posed by large-scale search spaces (Omidvar et al., 2021).

## 2.3 LEARNABLE EVOLUTIONARY SEARCH

Evolutionary optimization and incremental learning are innate methods humans employ to enhance their problem-solving capabilities (Michalski, 2000a). Relying solely on traditional evolutionary search strategies to solve LMOPs may be inadequate and inefficient (Wu et al., 2023), as the generator lacks the adaptability needed to grasp the precise characteristics of the LMOP they encounter (Bonissone et al., 2006). Consequently, it struggle to flexibly address the challenges posed by such black-box LMOPs. This is underscored by the fact that biological evolution can take thousands of

Figure 2: Illustration of the autoencoder-based learning and the innovization progress learning.

years to optimize a species (Miikkulainen & Forrest, 2021), whereas cumulative learning can dramatically accelerate this optimization process (Li et al., 2023). Moreover, the generator conducts iterative search of the variable space, generating a substantial dataset of feasible solutions. Employing machine learning (ML) techniques for the systematic analysis of these data enhances the understanding of search behavior and improves future search capabilities (Zhang et al., 2011).

Inspired by this, an intriguing research question emerges: Can we merge an evolutionary search with ML, creating learnable evolutionary search, to develop a more potent EA-based optimizer for efficiently addressing the scalability of LMOPs? Relevant existing attempts in this regard are given in the appendix A.1 and A.2. In an ideal scenario, a lightweight model $M(A)$ is trained using existing feasible solutions (i.e., data $D$) to enable one-shot or few-shot optimization. Precisely, after a generation or a few generations of evolutionary search, the trained model can directly output the target LMOP's Pareto optimal representation $x^*$ corresponding to each candidate solution $x$ in the current population. It can be expressed in the following mathematical form:

$$x^* = \Theta(x; A^*, \theta^*, D^*) \leftarrow (A^*, \theta^*, D^*) = \arg\min_D \{M(A), L(\theta)\} \qquad (2)$$

where three key components need to be identified for getting $x^*$: the well-prepared training data $D^*$, the lightweight architecture $A^*$, and the optimal model parameters $\theta^*$ to minimize the loss $L(\theta)$. Even if $x^*$ is not the Pareto optimal representation of $x$, its superior performance significantly contributes to accelerating the evolutionary optimization. Thus, rapid population convergence can be guaranteed theoretically. This is obviously a meaningful but very challenging multi-layer optimization problem. Nevertheless, this work seeks breakthroughs along this research direction to improve the performance and efficiency of EAs for solving complex LMOPs.

Similar initiatives include autoencoder-based learning (Tian et al., 2020), as depicted in Figure 2(a), which aims to obtain compressed representations in the code layer, and innovization progress learning (Mittal et al., 2021a), illustrated in Figure 2(b), which focuses on acquiring improvement representations. The autoencoder is primarily employed to reconstruct explored non-dominated solutions, lacking the ability to enhance solution quality, thus falling short in accelerating the convergence of the evolutionary search. The innovization progress model is mainly designed for repairing newly generated solutions (Mittal et al., 2021b), as indicated in formula (2), and may not fully exploit the potential of evolutionary search. Moreover, their reliance on relatively large models necessitates a substantial amount of training data, which can be inefficient and less adaptable as the optimization progresses. Typically, they draw data from extensive past populations. However, as the optimization progresses, the promising directions of improvement change, and past populations may mislead model training. Therefore, contemporary populations often provide a more accurate reflection of the path towards optimal future solutions. Building upon these insights, this study aims to train a lightweight MLP model that effectively leverages the current population. This trained model is then iteratively stacked to create a larger model, with the goal of capturing deep improvement representations of solutions. Subsequently, an evolutionary search is conducted within this learned representation space to maximize the potential for discovering high-quality solutions.

## 3 ACCELERATED EVOLUTIONARY OPTIMIZATION

The learnable MOEA (LMOEA) framework presented in this work closely resembles a standard MOEA, with the primary distinction residing in the generator component, as shown in Figure 1(b).

The pseudocode for the LMOEA process is given in the appendix, which consists of three fundamental steps: initialize a start parent population $P$ with $N$ random solutions, reproduce an offspring population $Q$ composed of $N$ child solutions by the generator, and filter half of the underperforming solutions from the combined population of $P + Q$ with the selector. This generator-selector iteration continues until a predefined stopping condition is met, typically when the total number of function evaluations reaches the maximum budget $FE_{max}$. What plays a major role in the generator is how to do effective evolutionary search. In this study, we design new learnable evolutionary search strategies in the learned representation space to acclerate the optimization for LMOPs.

## 3.1 BUILD A LIGHTWEIGHT MODEL

**Architecture $A^*$:** In our MLP design, both the input and output layers have the same number of neurons, aligning with the LMOP's variable size $(n)$. We've carefully considered the computational cost of integrating a ML model into an EA, opting for a single hidden layer with $K$ neurons to manage computational overhead (where $K << n$). The computational complexity of running this model is akin to traditional evolutionary search operators. The activation is the sigmoid function. Training the MLP involves iteratively updating its parameters (weights and biases) using backpropagation with gradient descent. Specifically, we calculate the steepest descent direction by evaluating the loss relative to the current parameters and iteratively adjust the parameters along this gradient descent direction to minimize the loss. For evaluation, the mean-square error (MSE) is used as the loss function to be minimized.

**Training Data $D^*$:** Given the training dataset $D = \left\{ \left( x_i, x_i^l \right) \right\}_{i=1}^{M}$, consisting of $M$ input-label examples, the goal is to adjust the MLP's parameters so that the actual output $y_i$ closely matches its corresponding label for all $i = 1, 2, \dots, M$, following statistical principles. The MLP undergoes supervised learning, guided by the labels $x^l$, with the ultimate expectation of acquiring knowledge about the performance improvement representation of a given input solution $x$. To ensure this representation is effective, it's essential that the label $x^l$ corresponds to a solution vector that surpasses $x$ according to predefined criteria. Furthermore, to ensure diversity within the dataset and encompass a broad range of scenarios for solving the target LMOP (i.e., generalization), we decompose it into $N$ subproblems, leveraging a set of uniformly distributed reference vectors $(r_1, r_2, \dots, r_N)$ in the objective space. The classical Penalty-based Boundary Intersection (PBI) approach is used to define each subproblem, which can be expressed mathematically as follows:

$$\text{Minimize } g\left(x \mid r_i\right) = d_1^i + d_2^i, \text{ where } d_1^i = F'(x)^T r_i / \left| r_i \right|, d_2^i = \left| F'(x) - \left( d_1^i / \left| r_i \right| \right) r_i \right| \quad (3)$$

PBI is a balanceable scalarizing function, which consists of two components, i.e., a convergence distance $d_1^i$ and a diversity distance $d_2^i$, where $d_1^i$ is the projection distance of $F'(x)$ on the $r_i$ and $d_2^i$ is the perpendicular distance between $F'(x)$ and $r_i$. The procedure for selecting an input-label pair of the $ith$ subproblem is as follows: Locate the two solutions from the current population $P$ with the smallest $d_2^i$, and designate the solution with the higher $g\left(x \mid r_i\right)$ value as the input $x$, with the other serving as its label $x^l$. Both objectives and variable values in the training data are normalized, with $x_i$ and $f_j(x)$ of solution $x$ normalized as follows:

$$\text{Normalization: } x_i' = \frac{x_i - L_i}{U_i - L_i}, i = 1, \dots, n; f_j'(x) = \frac{f_j(x) - z_j^{\min}}{z_j^{\max} - z_j^{\min}}, j = 1, \dots, m \quad (4)$$

where $z_j^{\min}$ and $z_j^{\max}$ are, respectively, the minimum and maximum values of the $ith$ objective for all solutions in $P$; $L_i$ and $U_i$ are the lowest and uppest bound of the $ith$ variable. These $N$ PBI subproblem-guided solution pairs form $D^*$. Thus, we start by initializing the MLP with random parameters and train it on $D^*$ using a learning rate of 0.1, momentum of 0.9, and 2 epochs.

## 3.2 DEEP ACCELERATED EVOLUTIONARY SEARCH

After training the MLP, new offspring of the target LMOP can be generated in four ways: 1) Traditional evolutionary search in the original space. 2) Inputting newly generated offspring into the MLP to obtain improvement representations directly. 3) Creating compressed representations, conducting an evolutionary search in the compressed space to generate new codes, and decoding them for improvement representations. 4) Obtaining improvement representations first and then evolutionary search in the improvement representation space. Expanding on the foundations laid by NSGA-II

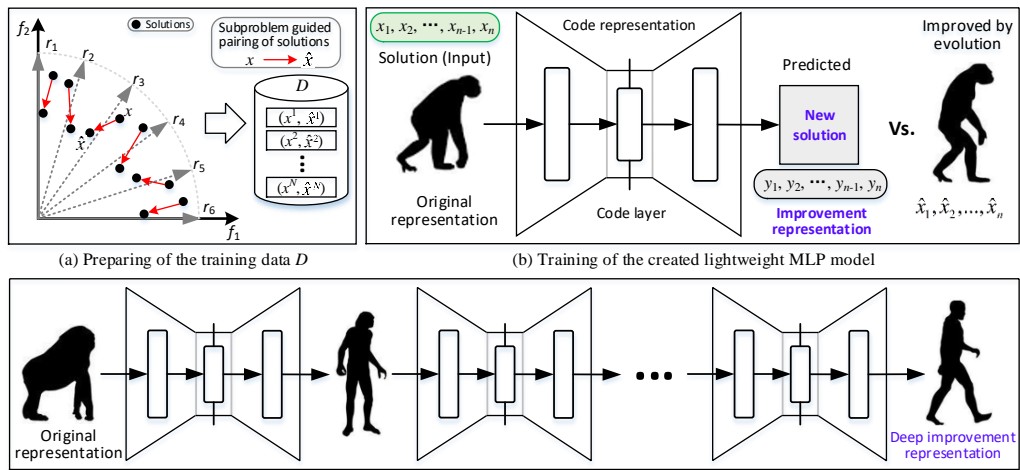

Figure 3: Illustration of the main process of the proposed deep improvement representation learning.

(Deb et al., 2002) and MOEA/D (Zhang & Li, 2007), we will delve into these four scenarios. In the first scenario, SBX and DE serve as the evolutionary search operators respectively in NSGA-II and MOEA/D. In the subsequent three scenarios, three distinct learnable MOEA variants are proposed for both NSGA-II (termed LNSGAV1-3) and MOEA/D (referred to as LMOEADV1-3). These variants improve upon the SBX and DE strategies by incorporating the MLP (see appendix A.3).

To further boost efficiency, we stack the trained MLP $t$ times to create a larger model. This expanded model provides a deeper improvement representation of solutions, as shown in Figure 3. Then, we can repair new generated solutions to get their DIRs or carry out evolutionary search within the DIR space, with the goal of substantially accelerating the optimization process and achieving few-shot optimization of LMOPs. Combining these two search strategies, another two new learnable MOEA variants for both NSGA-II (termed LNSGAV4-5) and MOEA/D (referred to as LMOEADV4-5) are developed. In addition, completely avoiding search in the original space carries the risk of losing crucial information, potentially leading to slow growth of the MLP model and a decline in overall optimization performance. To mitigate this concern, LNSGAV1-5 and LMOEADV1-5 balance between original and learnable evolutionary search with an adptive probability for each to generate offspring solutions at each generation. Their pseudo-code is provided in the appendix A.3.

## 4 EXPERIMENTAL STUDIES

The source codes for all the EA solvers and test LMOPs in our experimental studies are implemented on PlatEMO (Tian et al., 2023). We conduct all experiments on a personal computer with an Intel(R) Core(TM) i5-10505 CPU (3.2 GHz) and 24GB RAM. To ensure a statistically sound comparison, the proposed optimizers and their competitors run 20 times independently on each test problem. In each run, we set the termination condition as $FE_{max} = 10^5$. The population size $(N)$ is fixed at 100 for 2-objective LMOPs and 150 for 3-objective LMOPs. To assess the performance of an EA on LMOPs, we use two well-established metrics: inverted generational distance (IGD) (Ishibuchi et al., 2015) and hypervolume (HV) (Boelrijk et al., 2022). They gauge convergence and diversity in the final population. IGD is computed using $10^4$ points from the true Pareto front, while normalized HV employs a reference point $(1, 1, \ldots, 1)$. Smaller IGD and larger HV values signal better performance, indicating effective coverage of the true PF by the obtained final population.

### 4.1 EFFECTIVENESS VALIDATION OF PROPOSED ACCELERATED EVOLUTIONARY SEARCH

We commence the validation of the proposed accelerated evolutionary search strategies (NSGA-II vs. LNSGAV1-V5 and MOEA/D vs. LMOEADV1-V5) by optimizing synthetic LMOPs widely studied in the related literature. We focus on 2-objective DTLZ1 to DTLZ4 problems (Deb et al.),

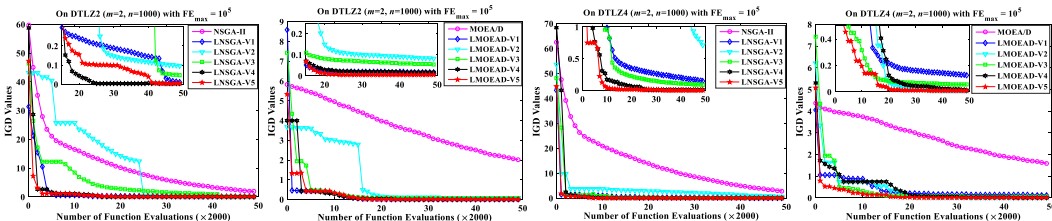

Figure 4: Illustration of the evolutionary process in solving DTLZ2 and DTLZ4 problems.

with the number of variables ($n$) varying from 1000 to 10000. The used MLP model's hidden layer consists of 10 neurons, and the MLP is stacked three times during the DIR learning process.

Figure 4 depicts the evolutionary process based on IGD results for comparisons involving 2-objective DTLZ2 and DTLZ4 problems with 1000 variables. These convergence graphs highlight the notable superiority of the improved versions (LNSGAV1-V5 and LMOEADV1-V5) over their respective original versions (NSGA-II and MOEA/D), particularly in terms of convergence speed. Specifically, when compared to NSGA-II (and likewise MOEA/D), most of its accelerated variants require only one-tenth of the computational resources to achieve near-Pareto optimal results for solving these two benchmarks. Furthermore, optimizers that explore the DIR space (LNSGAV4-5 and LMOEADV4-5) exhibit superior acceleration effects and final population performance.

Detailed IGD and HV results for solving 2-objective DTLZ1 to DTLZ4 problems with 1000 variables are given in Table 1, while the results for solving other DTLZ cases are presented in Tables 4 to 8 of the appendix. These results demonstrate the effectiveness of our proposed accelerated search strategies in improving evolutionary optimization efficiency. Nevertheless, several noteworthy observations can be drawn from these results: 1) The overall performance of all optimizers falls short when tackling DTLZ1 and DTLZ3, both of which are multimodal optimization problems, in which the number of local optima increases exponentially with the search space dimension. 2) The DIR-based search methods (LNSGAV4-5 and LMOEADV4-5) exhibit superior performance compared to their non-MLP stacking counterparts (LNSGAV1, LNSGAV3, LMOEADV1, and L-MOEADV3) in solving DTLZ2 and DTLZ4, but the results show the opposite trend for DTLZ1 and DTLZ3. 3) Solvers that rely on searching in the compressed representation space (LNSGAV2 and LMOEADV2) exhibit slightly less stability and are not as effective in accelerating convergence. 4) The learned model typically provides a short-term acceleration effect on evolutionary optimization, and its fundamental utility becomes less evident in the later stages of evolution.

Table 1: Average IGD and HV results of MOEA/D and its five accelerated versions on DTLZ1-4 with $m = 2, n = 1000, FE_{max} = 10^5$. The standard deviation indicated in parentheses following.

| Metric | Problem | MOEA/D | LMOEADV1 | LMOEADV2 | LMOEADV3 | LMOEADV4 | LMOEADV5 |
|---|---|---|---|---|---|---|---|
| IGD | DTLZ1 | 3.805e+3 | **1.114e+0** | 5.949e+0 | 4.947e+0 | 1.966e+1 | 5.903e+2 |
| | | (1.5e+3) | **(2.8e+0)** | (2.9e+2) | (1.9e+2) | (2.9e+2) | (1.8e+3) |
| | DTLZ2 | 1.945e+0 | 1.223e-2 | 8.074e-2 | 5.419e-2 | 1.127e-2 | **4.916e-3** |
| n=1000 | | (5.5e-1) | (1.1e-2) | (7.4e-2) | (6.2e-2) | (1.6e-1) | **(5.1e-3)** |
| | DTLZ3 | 1.172e+4 | **1.240e+1** | 3.047e+2 | 7.887e+2 | 1.273e+2 | 1.059e+3 |
| | | (3.6e+3) | **(2.6e+2)** | (8.3e+2) | (7.7e+2) | (6.8e+2) | (6.1e+3) |
| m=2 | DTLZ4 | 1.510e+0 | 1.288e-1 | 1.599e-2 | 5.569e-2 | 1.480e-2 | **8.609e-3** |
| | | (7.2e-2) | (1.3e-1) | (3.4e-1) | (8.9e-1) | (2.3e-2) | **(2.7e-2)** |
| HV | DTLZ1 | 0.00e+0 | **4.289e-2** | 1.605e-2 | 3.325e-2 | 0.00e+0 | 0.00e+0 |
| | | (0.0e+0) | **(1.0e-1)** | (1.1e-1) | (5.1e-1) | (0.0e+0) | (0.0e+0) |
| | DTLZ2 | 0.00e+0 | 3.340e-1 | 2.169e-1 | 2.583e-1 | 3.355e-1 | **3.506e-1** |
| n=1000 | | (0.0e+0) | (1.7e-2) | (1.4e-1) | (1.4e-1) | (1.2e-1) | **(1.7e-1)** |
| | DTLZ3 | 0.00e+0 | 0.00e+0 | 0.00e+0 | 0.00e+0 | 0.00e+0 | 0.000e+0 |
| | | (0.0e+0) | (0.0e+0) | (0.0e+0) | (0.0e+0) | (0.0e+0) | (0.0e+0) |
| m=2 | DTLZ4 | 0.00e+0 | 1.695e-1 | 3.026e-1 | 2.611e-1 | 3.174e-1 | **3.287e-1** |
| | | (0.0e+0) | (1.3e-1) | (1.5e-1) | (1.5e-1) | (1.5e-1) | **(2.0e-1)** |

There are several reasons for these observations. Firstly, the effectiveness of learning the improvement representation of solutions depends heavily on the quality of training data. Our training data

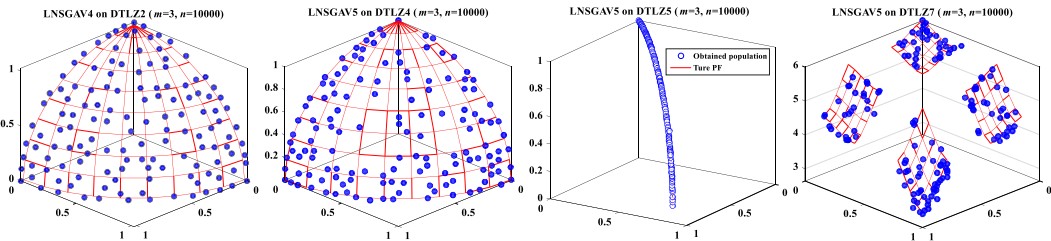

Figure 5: Illustration of the final solutions obtained by our proposed accelerated solvers on DTLZ2, DTLZ4, DTLZ5, and DTLZ7 with $m = 3, n = 10^4, FE_{max} = 10^5$.

is constructed based on how well solutions perform in the objective space. If there isn't a straightforward one-to-one correspondence between the search space and the objective space, such as in multi-modal problems, the learned MLP may not accurately capture the promising directions for improvement, and stacking pre-trained MLPs could potentially hinder the optimization process. Secondly, as the evolutionary process continues, the distinctions between different solutions tend to diminish, making the learned models progressively less helpful in aiding the optimization process.

## 4.2 COMPARISON WITH STATE-OF-THE-ART LMOEAS

To further evaluate the effectiveness of our DIR-based algorithms, namely LNSGAV4-V5 and LMOEADV4-5, we do a comparative analysis against five state-of-the-art LMOEAs (CCGDE3 (Antonio & Coello, 2013), LMOCSO (Tian et al., 2019), DGEA (He et al., 2020a), FDV (Yang et al., 2021), and MOEA/PSL (Tian et al., 2020)) representing different categories in solving 3-objective DTLZ1 to DTLZ7 problems. These competitors span a range of existing LMOEA approaches. The Table 9 in appendix contains the average IGD results for all considered solvers tackling these seven problems. These results clearly highlight the struggles most existing LMOEA competitors face when dealing with these large-scale DTLZ benchmarks. In contrast, our proposed optimizers, LNSGAV4-V5 and LMOEADV4-5, which employ deep accelerated evolutionary search with stacked MLP models, consistently outperform the five competitors when solving six out of seven DTLZ problems, although they do not achieve the best IGD results for DTLZ7. Additionally, Figure 5 illustrates the final solutions obtained by our algorithms for the $10^4$-dimensional DTLZ2, DTLZ4, DTLZ5, and DTLZ7 problems. These solutions (represented by blue points) closely approximate the true PF (red lines) of the target LMOP.

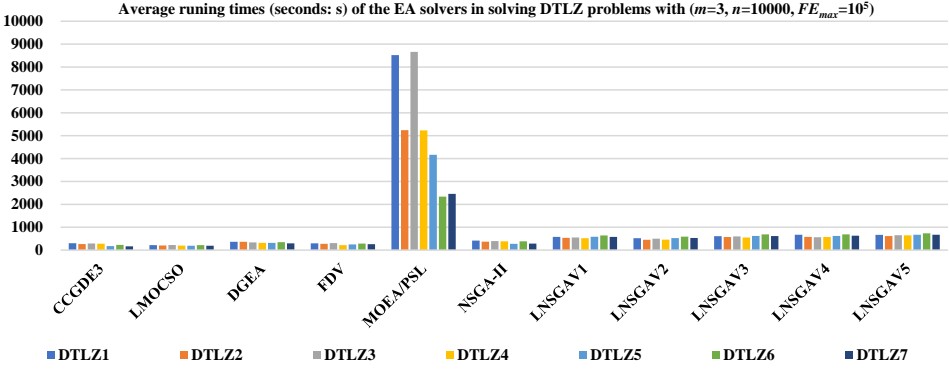

Figure 6: Illustration of the average running time (s) that each solver cost.

## 4.3 COMPARISON OF ACTUAL RUNNING TIMES

The practical runtimes of accelerated NSGA-II variants and their six competitors are evaluated for computational complexity. Figure 6 displays the average runtime (in seconds: s) for all ten optimizers over 20 runs on the 3-objective DTLZ1 to DTLZ7 problems with $n = 10^4, FE_{max} = 10^5$. No-

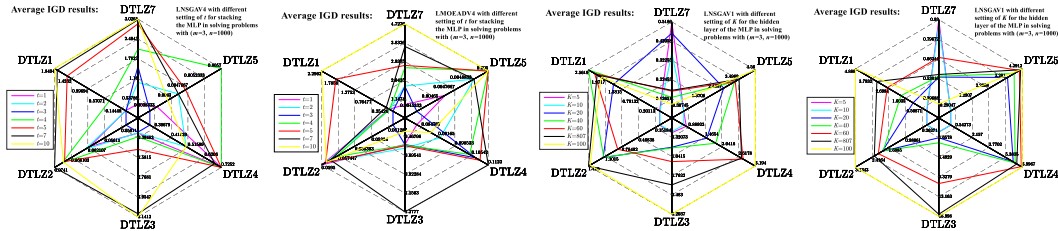

Figure 7: Illustration of the sensitivity analysis for two parameters $t$ and $K$.

Table 2: Average HV results of selected algorithms in solving real-world TREE problems

| Solvers | TREE1-3000 | TREE2-3000 | TREE3-6000 | TREE4-6000 | TREE5-6000 |
|---------|-----------|-----------|-----------|-----------|-----------|
| NSGAII | 6.095e-1(5.4e-3) | 6.691e-1(4.6e-3) | NaN(NaN) | NaN(NaN) | NaN(NaN) |
| MOEA/D | 7.523e-1(3.0e-3) | 7.788e-1(3.6e-3) | 7.268e-1(8.5e-3) | 1.045e-1(6.8e-2) | 6.807e-1(3.9e-3) |
| CCGDE3 | NaN(NaN) | NaN(NaN) | NaN(NaN) | NaN(NaN) | NaN(NaN) |
| LMOCSO | 8.063e-1(8.3e-3) | 7.876e-1(3.6e-3) | NaN(NaN) | 0.00e+0(0.0e+0) | NaN(NaN) |
| DGEA | 7.928e-1(3.6e-2) | 7.999e-1(1.2e-2) | 6.543e-1(2.6e-1) | 4.719e-1(4.0e-1) | 7.457e-1(2.4e-1) |
| FDV | 7.117e-1(5.0e-2) | 7.720e-1(4.8e-3) | NaN(NaN) | NaN(NaN) | NaN(NaN) |
| MOEA/PSL | 8.141e-1(1.7e-2) | 8.096e-1(5.3e-2) | 8.744e-1(2.3e-2) | 7.942e-1(1.86e-1) | 8.853e-1(5.19e-2) |
| LNSGAV5 | 8.115e-1(3.2e-2) | **8.34e-1(9.5e-2)** | 8.745e-1(1.5e-2) | 9.525e-1(1.9e-2) | 8.967e-1(2.3e-2) |
| LNSGAV6 | **8.36e-1(1.8e-2)** | 8.164e-1(3.9e-2) | **8.86e-1(1.5e-4)** | 9.212e-1(5.7e-2) | **9.21e-1(2.5e-3)** |
| LMEADV5 | 8.153e-1(5.9e-2) | 7.954e-1(4.3e-2) | 8.736e-1(1.6e-2) | **9.57e-1(2.8e-3)** | 8.834e-1(7.8e-2) |
| LMEADV6 | 7.824e-1(6.6e-2) | 8.058e-1(3.8e-2) | 8.828e-1(4.5e-3) | 9.021e-1(3.8e-1) | 9.116e-1(1.3e-2) |

tably, LNSGAV1 to LNSGAV5 exhibit similar runtimes to NSGA-II and most compared LMOEAs, suggesting that the lightweight MLP model's computational overhead in these learnable EAs is manageable. In contrast, MOEAPSL, utilizing a larger model and more training epochs, not only performs suboptimally but also incurs a higher computational cost. The underperformance of MOEA/PSL may also stem from its reliance on autoencoder-based learning, which limits its ability to acquire improvement representations of solutions.

### 4.4 PARAMETER SENSITIVITY ANALYSIS

We do sensitivity analysis on the number of stacked MLP models ($t$) for LNSGAV4 and LMOEAD-V4. Average IGD results in Figure 7 show that $t = 3$ yields best overall performance, with diminishing returns beyond this value. Additionally, we analyze the number of hidden layer nodes ($K$) in the MLP model for LNSGAV1 and LMOEADV1, revealing that $K = 5$ and $K = 10$ perform well, except for DTLZ7, where larger $K$ values more are advantageous. This is likely because lighter models are easier to train and perform better.

### 4.5 OPTIMIZATION OF REAL-WORLD LMOPS

We also tested our proposed algorithms on practical LMOPs, particularly the time-varying ratio error estimation (TREE) problems related to voltage transformers (He et al., 2020b). The results, summarized in Table 2, indicate that our algorithms with deep accelerated evolutionary search outperform the competitors across all five TREE problems in terms of HV scores.

## 5 CONCLUSIONS

This study proposes novel strategies to enhance evolutionary algorithms for LMOPs. Key contributions involve creating a lightweight model for learning improvement representations, assessing the impact of learnable evolutionary search, and designing a large model for deep improvement representation, all with the goal of efficient LMOP optimization. However, the method has limitations, including reliance on training data, limited effectiveness in multimodal problems, optimization instability, and short-term speed improvements.

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

# A  APPENDIX

We provide more discussion, details on the proposed methods, experimental results and analysis in this appendix.

## A.1  LEARNABLE MULTIOBJECTIVE EVOLUTIONARY ALGORITHMS

The conventional evolutionary generator and discriminator (or selector) in a typical MOEA are constructed using fixed genetic operators, such as crossover, mutation, and selection. Consequently, they lack the capability to learn and adapt to the specific characteristics of the optimization problem they are tasked with. As a result, they cannot effectively respond to the potential challenges posed by solving such a black-box optimization problem. In the context of evolutionary computation, research studies focused on learnable Multi-Objective Evolutionary Algorithms (MOEAs) have garnered substantial attention. Machine learning (ML) techniques are leveraged to support and enhance various modules within learnable MOEAs, including the generator, and discriminator (or selector).

Specifically, the MOEA's generator iteratively explores the variable space, generating a significant volume of data comprising feasible solutions. ML techniques systematically analyze this data to gain insights into the search behavior and enhance its future search capabilities. By traversing promising directions learned within the search space, the MOEA's generator efficiently identifies solutions with high potential (Michalski, 2000b; Michalski et al., 2007).

The MOEA's discriminator benefits from online predictions of Pareto Front (PF) shapes, enabling it to adeptly filter out underperforming solutions when dealing with MOPs featuring irregular PFs. Prior to tackling these problems, dimension reduction and spatial transformation techniques simplify both the objective and search spaces.

Furthermore, reinforcement learning (RL) techniques come into play in determining suitable evolutionary operators (i.e., actions) based on the current parent state. These actions guide the generator in producing high-quality offspring. Domain adaptation techniques are employed to learn domain-invariant feature representations across different optimization problems, enabling the analysis of distribution divergence. This knowledge transfer facilitates the sequential or simultaneous solution of these problems (Ma et al., 2021).

Bayesian optimization is an optimization technique that uses probabilistic models, like Gaussian Processes, to efficiently optimize complex, expensive functions. It sequentially samples the function, updates the surrogate model, and selects the next sampling point using an acquisition function (Khan et al., 2002). This process balances exploration and exploitation to find the global optimum. However, Bayesian optimization has limitations: it can be computationally expensive, struggles with high-dimensional spaces, assumes smooth functions, depends on the quality of the initial model, and may converge to local optima for multimodal functions.

## A.2  ONLINE INNOVIZATION OPERATORS

To enhance the search capability when addressing LMOPs, certain MOEAs have developed competitive learning-based search strategies. In these strategies, the population is divided into two groups: winners and losers, and the search process is guided so that the losers move closer to the winners. Typically, these competitions occur within the objective space. The effectiveness of these strategies heavily relies on the quality of the winners, as losers are guided towards them through competitive learning. If the winners encounter difficulties, such as falling into local optima, the entire evolutionary population may experience slow convergence. Furthermore, winners, often subjected to genetic operators, may only exhibit slight improvements. Consequently, the challenge lies in determining how these higher-quality winners can further evolve with a faster convergence rate.

To address this challenge and expedite the search process, efforts have been made through the design of online innovization operators (Deb & Srinivasan, 2006). The term "innovization" is derived from "innovation via optimization" and was originally defined as a post-optimality analysis of optimal solutions to provide valuable design principles to engineering designers. In the context of MOEAs, offspring generated by genetic operators are further refined by innovization operators to progress along the learned directions of performance improvement (Gaur & Deb, 2017). In the innovization process, various data-mining and machine learning techniques are employed to automatically uncov-

er innovative and crucial design principles within optimal solutions. These principles may include inter-variable relationships, commonalities among optimal solutions, and distinctions that set them apart from one another (Mittal et al., 2020). These operators enable the population to converge faster without consuming additional function evaluations compared to traditional local search methods.

Expanding on the concept of innovization, the notion of knowledge-driven optimization is introduced (Bandaru & Deb, 2010). In this approach, MOEAs assimilate knowledge, such as latent patterns shared by high-quality solutions, learned from intermediate solutions to guide their search away from mediocre solutions and toward potentially promising regions. This constitutes an online learning process that involves deciphering what makes one solution optimal (or near-optimal) within the final solution set and understanding what causes one solution to outperform another during the optimization process. Online innovization aims to accelerate evolutionary search and enhance the efficiency of generators.

In an online innovization operator, a supervised learning model is typically constructed and trained online with the aim of implicitly learning historical directional improvements, such as transitioning from dominated to non-dominated solutions, within the previously explored search space. A solution $x = (x_1, x_2, \ldots, x_n)$ is identified as performing poorly, while $x^*$ represents a high-quality solution. Throughout the evolutionary process, various solution pairs $(x, x^*)$ can be collected. Subsequently, the selected model, which can be a multilayer perceptron, a random forest, or a deep neural network, is trained using this labeled data. In this context, $x$ serves as the input, and $x^*$ serves as its label or expected target output. The trained model is believed to have the capacity to capture underlying patterns that reflect the directional improvement of solutions within the search space.

Ideally, a newly generated offspring solution, $x^{new}$, produced by genetic operators can be enhanced (or progressed) by inputting it into the well-trained model to obtain an improved version, $y^{new}$, as the output. This process of repairing offspring is also referred to as innovized progress. It holds the potential to enhance the search capability of generators when tackling scalable MOPs, primarily due to the following four merits: 1) Incorporation of all conflicting objective information during the learning process. 2) Elimination of the need for additional function evaluations during the innovized progress of solutions. 3) Adaptability of the learned directional improvement of solutions as generations progress. 4) The potential for a substantial leap in the objective space when transitioning from $x^{new}$ to $y^{new}$ in the search space, which can expedite the search process. However, it's crucial to consider four key considerations when customizing such an innovization progress:

**Selection of the learning model:** The choice of the learning model is flexible and can align with available supervised learning models that meet the requirements. However, it's essential to take into account the computational cost associated with training the selected model.

**Collection of training data:** The process of collecting training data involves gathering paired data based on the performance of available solutions, either from previous generations or the current one. Therefore, when selecting a pair $(x, x^*)$, it is crucial to consider that $x^*$ outperforms $x$. For example, in a Pareto-based MOEA, $x^*$ should dominate $x$, in a decomposition-based MOEA, the aggregation fitness of $x^*$ should be superior to that of $x$, or $x^*$ should have the potential to guide $x$ towards rapid convergence.

**Training of the adopted model:** The process of training a model is itself an optimization problem, involving numerous hyperparameters such as model architecture, learning rate, and training epochs, which often require manual tuning. This can lead to various challenges, including the risk of overfitting the model. Additionally, it's essential to investigate whether the model should be updated regularly, such as every generation, and whether training should be conducted online or offline.

**Advancement of the search capability with the learned model:** The expectation is that the generator's search capability can be enhanced with the assistance of this well-trained model. Specifically, subpar and average solutions within the population can be repaired, facilitating rapid convergence in the learned promising direction. Simultaneously, high-quality solutions can be further improved to explore a broader range of elite solutions. However, two important considerations arise: Is it necessary to repair all newly generated solutions? Is it necessary to perform the innovization progress in every generation?

**Autoencoder-based representation learning:** The concept of representation learning has been previously introduced in MOEA/PSL, where it focuses on acquiring a compressed representation of the

input solution, referred to as Pareto Subspace Learning (PSL). This approach involves training a Denoise Autoencoder (DAE), a specific type of artificial neural network with equal nodes in the input and output layers. The DAE comprises two main components: an encoder and a decoder. The encoder learns to create a representation (or code) of the partially corrupted input, while the decoder maps this representation back to a reconstructed input. Through this process, the DAE can extract higher-level features from the input distribution.

During training, the DAE iteratively minimizes the reconstruction error, which measures the disparity between the output and the input. This training process resembles that of a feedforward neural network. To elaborate, each input undergoes perturbation via the mutation operator, and the non-dominated solutions within the current population serve as the training data.

Following training, each solution can be mapped between the original search space and the code representation space. Subsequently, a new offspring, denoted as $y$, can be generated through the following process:

**Step 1:** select two random parents $x^1$ and $x^2$ from the current population.

**Step 2:** map $x^1$ and $x^2$ to the code space to get their corresponding representations $c^1$ and $c^2$. The value of each $c_i^1 \in c^1$ (the same as $c^1$) in the code layer can be computed as follows,

$$c_i = \sigma \left( b_i + \sum_i x_i w_{ij} \right) \tag{5}$$

**Step 3:** run evolutionary search (e.g., SBX, DE, PM) on $c^1$ and $c^2$ to generate a new code $c$;

**Step 4:** map $c$ back to the original space for getting the new solution $y$ and the value of each $y_j \in y$ can be computed by

$$y_j = \sigma \left( b_j' + \sum_j c_i w_{ji}' \right) \tag{6}$$

where $b$ and $w$ are respectively the bias and the weight of this DAE with only one hidden layer, while $\sigma$ represents the sigmoid function.

### A.3 PSEUDOCODE FOR THIS TO WORK

Here is the pseudo-code of the algorithms designed in this paper, including the general MOEA algorithm framework and our two proposed improved learnable MOEA algorithm frameworks.

**Algorithm 1** is the basic MOEA algorithm framework. It differs from Algorithms 2 and 3 in the way solutions are generated.

**Algorithm 2** corresponds to our proposed algorithms LNSGAV1, LNSGAV4, LMOEADV1, LMOEADV4. **Algorithm 3** corresponds to our proposed algorithms LNSGAV2, LNSGAV3, LNSGAV5, LMOEADV2, LMOEADV3, LMOEADV5. Among these variants, LNSGAV2 and LMOEADV2 utilize the learned compressed representation space for searching, employing SBX and DE, respectively. Variants (LNSGAV1, LNSGAV3) and (LMOEADV1, LMOEADV3) leverage the learned improvement representation space for their search operations, employing SBX and DE, respectively. Finally, variants (LNSGAV4, LNSGAV5) and (LMOEADV4, LMOEADV5) utilize the learned deep improvement representation space for their search, employing SBX and DE, respectively. The detailed configuration is listed in Table 3.

In this study, the process of using SBX to generate a child solution is as follows:

**Step 1:** Randomly select two different parent solutions: $x^1 = \left( x_1^1, \cdots, x_n^1 \right)$ and $x^2 = \left( x_1^2, \cdots, x_n^2 \right)$ from the current population $P$;

**Step 2:** generate a child solution $c = (c_1, \cdots, c_n)$, where $c_i$ is computed as follows:

$$c_i = 0.5 \times \left[ (1 + \beta) \cdot x_i^1 + (1 - \beta) \cdot x_i^2 \right] \tag{7}$$

where $\beta$ is dynamically computed as follows:

$$\beta = \begin{cases} (\text{ rand } \times 2)^{1/(1+\eta)} & \text{rand } \leq 0.5 \\ (1/(2 - \text{ rand } \times 2))^{1/(1+\eta)} & \text{otherwise.} \end{cases} \tag{8}$$

where $\eta$ is a hyperparameter (the spread factor distribution index), which is set as 20. The greater the value of $\eta$, the greater the probability that the resulting child solution will be close to the parent.

The DE/rand/1 operator is used in this study. For each solution $x \in P$, the process of using DE to generate a child solution of $x$ is as follows:

**Step 1:** Pick two solutions $x^1$ and $x^2$ from the population $P$ at random, they must be distinct from each other as well as from the base vector $x$.

**Step 2:** The mutated individual $v$ is obtained according to the following formula:

$$v_i = x_i + F(x_i^1 - x_i^2) \tag{9}$$

**Step 3:** The final individual $c$ is obtained by crossover according to the following formula

$$c_i = \begin{cases} v_i & \text{if } \text{rand}_i[0,1] \leq CR \text{ or } i = k \\ x_i & \text{Otherwise} \end{cases} \tag{10}$$

In this study, we set $F = 0.5$ and $CR = 0.75$.

---

**Algorithm 1:** The general framework of an MOEA

---

**Input:** the LMOP with $m$ objectives and $n$ variables, the function evaluation budget $FE_{max}$
**Output:** the final population $P$ to approximate the PF/PS
initialize $P$ with $N$ random solutions;
initialize the function evaluation counter $FE = 0$;
**while** $FE \leq FE_{max}$ **do**
    $Q$ = Generator($P$); *//evolutionary search in variable space to find new offspring solutions.*
    $P$ = Selector(P, Q); *//environmental selection in objective space to filter poor solutions.*
    $FE = FE + N$;
**end**
**return** $P$

---

**Algorithm 2:** The general framework of the proposed LMOEA-V1

---

**Input:** the LMOP with $m$ objectives and $n$ variables, the function evaluation budget $FE_{max}$
**Output:** the final population $P$ to approximate the PF/PS
initialize $P$ with $N$ random solutions;
initialize a set of $N$ uniformly distributed reference vectors $R = (r_1, r_2, \ldots, r_N)$;
initialize the function evaluation counter $FE = 0$;
**while** $FE \leq FE_{max}$ **do**
    initialize an MLP model $M(A^*)$ with random parameters;
    $D^*$ = TrainingDataPreparation(P, R); *//PBI subproblem-guided pairing of solutions.*
    update the parameters of $M(A^*)$ via backpropagation with gradient descent;
    **for** $i = 1$ *to* $N$ **do**
        search in the original variable space to generate an offspring solution;
        **if** $rand > FE/FE_{max}$ **then**
            repair the new generated solution by $M(A^*)$ to be its improvement representation;
        **end**
        add the new generated offspring solution into $Q$;
    **end**
    $P$ = Selector(P, Q); *//environmental selection in the objective space.*
    $FE = FE + N$;
**end**
**return** $P$

---

## A.4 TIME-VARYING RATIO ERROR ESTIMATION

The precise estimation of voltage transformers' (VTs) ratio error (RE) holds significant importance in modern power delivery systems. Existing RE estimation methods predominantly revolve around

---

**Algorithm 3:** The general framework of the proposed LMOEA-V2 to LMOEA-V5

---

**Input:** the LMOP with $m$ objectives and $n$ variables, the function evaluation budget $FE_{max}$
**Output:** the final population $P$ to approximate the PF/PS
initialize $P$ with $N$ random solutions;
initialize a set of $N$ uniformly distributed reference vectors $R = (r_1, r_2, \ldots, r_N)$;
initialize the function evaluation counter $FE = 0$;
**while** $FE \leq FE_{max}$ **do**
    initialize an MLP model $M(A^*)$ with random parameters;
    $D^* =$ TrainingDataPreparation(P, R); *//PBI subproblem-guided pairing of solutions.*
    update the parameters of $M(A^*)$ via backpropagation with gradient descent;
    **for** $i = 1$ *to* $N$ **do**
        **if** $rand < FE/FE_{max}$ **then**
            search in the original variable space to generate an offspring solution;
        **end**
        **else**
            search in the learned representation space by $M(A^*)$ to generate a solution;
        **end**
        add the new generated offspring solution into $Q$;
    **end**
    $P =$ Selector(P, Q); *//environmental selection in the objective space.*
    $FE = FE + N$;
**end**
**return** $P$

---

Table 3: The search and selection strategy configuration of our proposed algorithms

| Algorithms | Evolutionary search of the Generator | Environmental selection |
|---|---|---|
| NSGA-II | SBX in the original space | Pareto-based selection |
| LNSGA-V1 | SBX-based search in the original space followed by reparing part of offspring with MLP | Pareto-based selection |
| LNSGA-V2 | SBX-based search in the original space + SBX in the compressed representation space | Pareto-based selection |
| LNSGA-V3 | SBX-based search in the original space + SBX in the improvement representation space | Pareto-based selection |
| LNSGA-V4 | SBX-based search in the original space followed by repairing part of offspring with stacked MLP | Pareto-based selection |
| LNSGA-V5 | SBX-based search in the original space + SBX in the deep improvement representation space | Pareto-based selection |
| MOEA/D | DE in the original space | Decomposition-based selection |
| LMOEAD-V1 | DE-based search in the original space followed by reparing part of offspring with MLP | Decomposition-based selection |
| LMOEAD-V2 | DE-based search in the original space + DE in the compressed representation space | Decomposition-based selection |
| LMOEAD-V3 | DE-based search in the original space + DE in the improvement representation space | Decomposition-based selection |
| LMOEAD-V4 | DE-based search in the original space followed by repairing part of offspring with stacked MLP | Decomposition-based selection |
| LMOEAD-V5 | DE-based search in the original space + DE in the deep improvement representation space | Decomposition-based selection |

periodic calibration, disregarding the time-varying aspect. This oversight presents challenges in achieving real-time VT state estimation. To address this concern, the formulation of a time-varying RE estimation (TREE) problem as a large-scale multiobjective optimization problem is proposed in (He et al., 2020b). Multiple objectives and inequality constraints are defined based on statistical and physical rules extracted from power delivery systems. Additionally, a benchmark test suite is

Table 4: Average IGD and HV results of NSGA-II and its five accelerated versions on DTLZ1-4 with $m = 2, n = 1000, FE_{max} = 10^5$. The standard deviation indicated in parentheses following.

| Metric | Problem | NSGA-II | LNSGA-V1 | LNSGA-V2 | LNSGA-V3 | LNSGA-V4 | LNSGA-V5 |
|---|---|---|---|---|---|---|---|
| IGD | DTLZ1 | 4.477e+3 (5.3e+1) | **7.366e+0** **(9.6e+0)** | 6.010e+1 (5.8e+1) | 4.143e+2 (1.2e+2) | 1.957e+1 (2.0e+1) | 1.913e+3 (5.3e+3) |
| | DTLZ2 | 2.004e+0 (1.8e-1) | 1.291e-2 (4.7e-2) | 9.554e-2 (3.2e-2) | 4.820e-2 (2.6e-2) | 4.985e-3 (7.8e-3) | **4.849e-3** **(3.0e-3)** |
| n=1000 | DTLZ3 | 1.144e+4 (3.9e+2) | **7.236e-1** **(7.6e-1)** | **5.786e-1** **(4.5e-1)** | 1.767e+2 (1.2e+2) | 2.453e+2 (7.4e+2) | 1.258e+3 (5.0e+3) |
| m=2 | DTLZ4 | 2.935e+0 (1.2e-1) | 1.652e-1 (3.4e-1) | 7.123e-1 (2.9e-1) | 8.984e-2 (1.9e-1) | 1.286e-2 (2.6e-2) | **6.248e-3** **(8.4e-3)** |
| HV | DTLZ1 | 0.00e+0 (0.0e+0) | **2.938e-1** **(3.0e-1)** | 0.00e+0 (0.0e+0) | 0.00e+0 (0.0e+0) | 0.00e+0 (0.0e+0) | 0.00e+0 (0.0e+0) |
| | DTLZ2 | 0.00e+0 (0.0e+0) | 3.230e-1 (1.6e-1) | 1.830e-1 (4.6e-2) | 2.760e-1 (1.4e-1) | 3.429e-1 (1.2e-1) | **3.457e-1** **(1.0e-1)** |
| n=1000 | DTLZ3 | 0.00e+0 (0.0e+0) | 9.807e-2 (4.4e-2) | **1.197e-1** **(6.3e-1)** | 0.00e+0 (0.0e+0) | 0.00e+0 (0.0e+0) | 0.00e+0 (0.0e+0) |
| m=2 | DTLZ4 | 0.00e+0 (0.0e+0) | 1.716e-1 (1.2e-1) | 9.751e-2 (1.2e-1) | 2.134e-1 (1.4e-1) | 1.659e-1 (1.4e-1) | **3.046e-1** **(1.4e-1)** |

systematically created, encompassing various TREE problems from different substations to depict their distinct characteristics. This formulation not only transforms a costly RE estimation task into a more economical optimization problem but also contributes to the advancement of research in large-scale multiobjective optimization by providing a real-world benchmark test suite featuring intricate variable interactions and objective correlations. The source code for these optimization problems can be found on the PlatEMO.

## A.5 SUPPLEMENTARY EXPERIMENTAL STUDIES

Due to space limitations, a supplement of some experimental data from this work is provided here. Mainly are the average IGD and HV results of each algorithm in solving the DTLZ problem with different settings.

Table 5: Average IGD and HV results of NSGA-II and its five accelerated versions on DTLZ1-4 with $m = 2, n = 5000, FE_{max} = 10^5$. The standard deviation indicated in parentheses following.

| Metric | Problem | NSGA-II | LNSGA-V1 | LNSGA-V2 | LNSGA-V3 | LNSGA-V4 | LNSGA-V5 |
|---|---|---|---|---|---|---|---|
| IGD | DTLZ1 | 7.274e+4 (1.7e+3) | **1.823e+0** **(4.4e+0)** | 5.372e+0 (7.2e+0) | 1.149e+2 (9.3e+3) | 1.647e+2 (4.0e+3) | 2.706e+2 (3.3e+3) |
| | DTLZ2 | 1.514e+2 (6.3e+0) | 1.919e-2 (3.2e-2) | 7.376e+0 (1.7e+1) | 1.403e-2 (2.9e-2) | 1.017e-2 (4.0e-2) | **8.503e-3** **(6.2e-3)** |
| n=5000 | DTLZ3 | 1.931e+5 (6.6e+3) | **3.447e+3** **(8.0e+3)** | **9.285e+3** **(2.0e+4)** | 1.261e+4 (2.5e+4) | 4.499e+3 (9.5e+3) | 3.675e+3 (8.4e+3) |
| m=2 | DTLZ4 | 1.712e+2 (1.1e+1) | 2.107e-1 (2.9e-1) | 2.799e-1 (3.7e-1) | 1.304e-1 (2.0e-1) | 1.980e-1 (2.5e-1) | **7.511e-2** **(1.8e-1)** |
| HV | DTLZ1 | 0.00e+0 (0.0e+0) | **4.607e-1** **(2.3e-1)** | 1.916e-1 (2.9e-1) | 0.00e+0 (0.0e+0) | 0.00e+0 (0.0e+0) | 0.00e+0 (0.0e+0) |
| | DTLZ2 | 0.00e+0 (0.0e+0) | 3.253e-1 (4.6e-2) | 2.302e-1 (1.7e-1) | 3.150e-1 (1.7e-1) | 3.264e-1 (1.7e-1) | **3.465e-1** **(1.6e-1)** |
| n=5000 | DTLZ3 | 0.00e+0 (0.0e+0) | 1.153e-1 (1.7e-1) | **5.784e-2** **(1.4e-1)** | 1.153e-1 (1.7e-1) | 9.413e-2 (1.8e-1) | 1.141e-1 (1.3e-1) |
| m=2 | DTLZ4 | 0.00e+0 (0.0e+0) | 2.181e-1 (1.4e-1) | 2.198e-1 (1.4e-1) | 2.692e-1 (1.8e-1) | 2.472e-1 (1.3e-1) | **2.851e-1** **(1.4e-1)** |

### A.5.1 FUTURE RESEARCH DIRECTIONS

**Enhancing Evolutionary Selectors or Discriminators Through Machine Learning:** In the context of Many-Objective Optimization Problems (MaOPs), the application of machine learning tech-

niques serves as a subtle yet powerful augmentation to the environmental selection process. This augmentation proves invaluable when confronted with the escalating complexity of objective spaces within MaOPs. As the number of objectives in MaOPs increases, the efficacy of traditional environmental selection strategies in distinguishing subpar solutions from elite ones diminishes significantly. More precisely, the convergent pressure of the discriminator falls short, and its ability to maintain solution diversity becomes inadequate. Consequently, it becomes imperative to explore methods for augmenting the discriminative capabilities of environmental selection strategies when tackling MaOPs.

**Empowering Evolutionary Generators with Machine Learning:** Within the realm of Large-Scale Multi-Objective Problems (LMOPs), the integration of machine learning techniques plays a strategic role in augmenting the evolutionary search process. This augmentation enables a dynamic response to the formidable challenges posed by the expansive search spaces characteristic of LMOPs. In such vast search spaces, the effectiveness of conventional genetic operators markedly declines, resulting in the unfortunate consequence of generating suboptimal offspring by the generator. Hence, it becomes imperative to delve into methods aimed at elevating the search prowess of these generators when tackling LMOPs.

**Advancing Evolutionary Modules Through Learnable Transfer Techniques:** In the realm of multi-objective optimization problems (MOPs), we introduce evolutionary modules employing transfer learning principles to facilitate the exchange of valuable optimization insights between source and target MOPs. In essence, this approach serves as a shortcut to solving target MOPs by leveraging the knowledge acquired from the optimization processes of related source problems. The optimization of source MOPs can be accomplished either concurrently with or prior to addressing the target MOPs, leading to two distinct forms of transfer optimization: sequential and multitasking. In the sequential form, the target MOPs are tackled one after another, benefiting from the cumulative wisdom gleaned from prior optimization exercises. This approach ensures that the experiences garnered from solving earlier problems are effectively applied to optimize subsequent ones. In contrast, the multitasking form involves the simultaneous optimization of all MOPs from the outset, with each problem drawing upon the knowledge cultivated during the optimization of other MOPs. This collaborative optimization approach maximizes the utility of learned knowledge, significantly enhancing the efficiency of solving multiple MOPs simultaneously.

Table 6: Average IGD and HV results of NSGA-II and its five accelerated versions on DTLZ1-4 with $m = 2, n = 10^4, FE_{max} = 10^5$. The standard deviation indicated in parentheses following.

| Metric | Problem | NSGA-II | LNSGA-V1 | LNSGA-V2 | LNSGA-V3 | LNSGA-V4 | LNSGA-V5 |
|---|---|---|---|---|---|---|---|
| IGD | DTLZ1 | 1.993e+5 | **3.055e+1** | **2.068e+1** | 7.163e+1 | 4.098e+1 | 4.805e+1 |
| | | (3.8e+3) | **(6.8e+1)** | **(2.3e+1)** | (1.5e+2) | (9.7e+1) | (8.5e+1) |
| n= | DTLZ2 | 4.251e+2 | 8.651e-3 | 9.888e-3 | 1.481e-2 | 8.039e-3 | **7.789e-3** |
| 10000 | | (8.1e+0) | (8.7e-3) | (6.2e-3) | (4.2e-2) | (8.2e-3) | **(6.2e-3)** |
| | DTLZ3 | 5.616e+5 | **5.040e+0** | **6.319e+2** | 1.415e+2 | 1.115e+3 | 2.994e+2 |
| m=2 | | (8.7e+3) | **(1.1e+1)** | **(1.5e+3)** | (1.6e+2) | (2.7e+3) | (4.6e+3) |
| | DTLZ4 | 4.414e+2 | 1.686e-1 | 2.682e-1 | 1.060e-1 | 6.481e-2 | **1.045e-2** |
| | | (5.7e+0) | (2.9e-1) | (4.1e-1) | (2.5e-1) | (1.0e-2) | **(1.4e-2)** |
| HV | DTLZ1 | 0.00e+0 | **1.934e-1** | 8.803e-2 | 7.176e-2 | **3.269e-1** | 1.514e-1 |
| | | (0.0e+0) | **(3.0e-1)** | (2.1e-1) | (1.7e-1) | **(2.8e-1)** | (2.4e-1) |
| n= | DTLZ2 | 0.00e+0 | 3.407e-1 | 3.425e-1 | 3.152e-1 | 3.385e-1 | **3.438e-1** |
| 10000 | | (0.0e+0) | (1.3e-2) | (4.4e-3) | (1.9e-1) | (1.2e-2) | **(1.6e-1)** |
| | DTLZ3 | 0.00e+0 | **1.762e-1** | **6.261e-2** | 0.00e+0 | 0.00e+0 | 5.786e-2 |
| m=2 | | (0.0e+0) | **(1.7e-1)** | **(1.4e-1)** | (0.0e+0) | (0.0e+0) | (1.4e-1) |
| | DTLZ4 | 0.00e+0 | 2.598e-1 | 2.306e-1 | 2.645e-1 | 2.820e-1 | **3.076e-1** |
| | | (0.0e+0) | (1.3e-1) | (1.7e-1) | (1.1e-1) | (1.8e-1) | **(1.6e-1)** |

Table 7: Average IGD and HV results of MOEA/D and its five accelerated versions on DTLZ1-4 with $m = 2, n = 5000, FE_{max} = 10^5$. The standard deviation indicated in parentheses following.

| Metric | Problem | MOEA/D | LMOEADV1 | LMOEADV2 | LMOEADV3 | LMOEADV4 | LMOEADV5 |
|---|---|---|---|---|---|---|---|
| IGD | DTLZ1 | 2.645e+4 | **3.449e+0** | 5.419e+2 | 1.126e+2 | 1.072e+2 | 5.725e+2 |
| | | (8.4e+3) | **(7.5e+0)** | (1.3e+2) | (1.2e+2) | (2.6e+2) | (1.3e+3) |
| | DTLZ2 | 2.106e+1 | 9.524e-3 | 8.157e-3 | 1.151e-2 | 6.612e-3 | **6.442e-3** |
| n=5000 | | (4.1e+0) | (7.7e-4) | (1.8e-4) | (1.9e-2) | (6.4e-3) | **(4.9e-3)** |
| | DTLZ3 | 7.016e+4 | 1.703e+2 | **4.303e+1** | 3.896e+3 | 2.942e+3 | 3.054e+3 |
| | | (1.1e+4) | (4.1e+2) | **(5.3e+1)** | (4.5e+3) | (4.4e+3) | (4.7e+3) |
| m=2 | DTLZ4 | 1.433e+1 | 5.325e-1 | 5.145e+0 | 4.159e-1 | 3.105e-1 | **2.053e-1** |
| | | (3.5e+0) | (6.6e-1) | (8.6e+0) | (1.7e+0) | (3.5e-1) | **(4.4e-1)** |
| HV | DTLZ1 | 0.00e+0 | **2.036e-1** | 0.00e+0 | 0.00e+0 | 9.560e-2 | 1.508e-2 |
| | | (0.0e+0) | **(1.3e-1)** | (0.0e+0) | (0.0e+0) | (1.4e-1) | (3.7e-2) |
| | DTLZ2 | 0.00e+0 | 3.041e-1 | 3.149e-1 | 2.269e-1 | **3.471e-1** | 3.449e-1 |
| n=5000 | | (0.0e+0) | (1.6e-2) | (4.9e-2) | (1.7e-1) | **(8.4e-2)** | (1.7e-1) |
| | DTLZ3 | 0.00e+0 | **8.548e-2** | 2.126e-2 | 0.00e+0 | 4.454e-2 | 0.00e+0 |
| | | (0.0e+0) | **(6.2e-2)** | (5.1e-2) | (0.0e+0) | (6.4e-2) | (0.0e+0) |
| m=2 | DTLZ4 | 0.00e+0 | 1.996e-1 | 2.153e-1 | 2.670e-1 | 2.749e-1 | **2.815e-1** |
| | | (0.0e+0) | (1.5e-1) | (1.7e-1) | (1.3e-1) | (1.4e-1) | **(1.2e-1)** |

Table 8: Average IGD and HV results of MOEA/D and its five accelerated versions on DTLZ1-4 with $m = 2, n = 10^4, FE_{max} = 10^5$. The standard deviation indicated in parentheses following.

| Metric | Problem | MOEA/D | LMOEADV1 | LMOEADV2 | LMOEADV3 | LMOEADV4 | LMOEADV5 |
|---|---|---|---|---|---|---|---|
| IGD | DTLZ1 | 5.122e+4 | **1.692e-1** | 1.670e+0 | 6.430e+1 | 1.294e+1 | 2.858e+1 |
| | | (1.1e+4) | **(1.8e-2)** | (3.4e+0) | (1.2e+2) | (2.3e+1) | (3.1e+1) |
| n= | DTLZ2 | 4.815e+1 | **5.108e-3** | 6.582e-2 | 4.342e-2 | 2.127e-2 | 7.635e-3 |
| 10000 | | (6.8e+0) | **(2.2e-4)** | (1.4e-1) | (1.0e-2) | (3.2e-1) | (9.5e-3) |
| | DTLZ3 | 1.517e+5 | 2.812e+1 | 3.389e+2 | **2.201e+1** | 6.353e+2 | 5.255e+2 |
| | | (6.2e+4) | (6.7e+1) | (8.3e+2) | **(6.5e+1)** | (8.6e+2) | (8.2e+2) |
| m=2 | DTLZ4 | 3.573e+1 | 3.554e-1 | 1.276e+0 | 2.646e-1 | 3.396e-1 | **1.796e-1** |
| | | (3.3e+0) | (3.7e-1) | (1.2e+0) | (7.3e-1) | (4.9e-1) | **(4.0e-1)** |
| HV | DTLZ1 | 0.00e+0 | **3.050e-1** | 2.452e-1 | 1.434e-1 | 1.384e-1 | 1.909e-1 |
| | | (0.0e+0) | **(1.7e-2)** | (2.9e-1) | (2.4e-1) | (1.5e-1) | (2.2e-1) |
| n= | DTLZ2 | 0.00e+0 | **3.452e-1** | 2.919e-1 | 2.840e-1 | 3.008e-1 | 3.229e-1 |
| 10000 | | (0.0e+0) | **(2.3e-4)** | (1.2e-1) | (1.3e-1) | (1.7e-1) | (1.7e-1) |
| | DTLZ3 | 0.00e+0 | 1.062e-1 | 4.207e-2 | **1.209e-1** | 2.087e-2 | 6.192e-2 |
| | | (0.0e+0) | (5.2e-2) | (6.5e-2) | **(1.2e-1)** | (5.1e-2) | (6.8e-2) |
| m=2 | DTLZ4 | 0.00e+0 | 2.347e-1 | 1.278e-1 | 2.703e-1 | 2.360e-1 | **2.713e-1** |
| | | (0.0e+0) | (1.3e-1) | (1.6e-1) | (1.3e-1) | (1.5e-1) | **(1.7e-1)** |

Table 9: Average IGD results of deep accelerated LNSGAV4-5, LMOEADV4-5 and their five state-of-the-art LMOEA competitors on DTLZ1 to DTLZ7 with $m = 3$, $n \in (1000, 5000, 10000)$, $FE_{max} = 10^5$. The standard deviation indicated in parentheses following.

| Problems | n | CCGDE3 | LMOCSO | DGEA | FDV | MOEAPSL | LNSGAV4 | LNSGAV5 | LMOEADV4 | LMOEADV5 |
|---|---|---|---|---|---|---|---|---|---|---|
| DTLZ1 | 1000 | 1.7217e+4(1.00e+3) | 3.2312e+3(7.24e+2) | 2.3344e+3(1.53e+3) | 1.6134e+2(8.56e+0) | 7.2991e+3(5.32e+1) | 1.5451e+2(1.64e+2) | 3.8213e+1(1.00e+2) | **3.2785e+1(5.45e+1)** | 8.3810e+1(1.88e+2) |
|  | 5000 | 9.0465e+4(6.36e+3) | 1.5374e+4(3.71e+3) | 1.3147e+4(3.22e+3) | 1.1790e+3(1.85e+2) | 3.6479e+4(3.18e+2) | 4.4894e-1(8.61e-1) | 3.1124e-1(2.44e-1) | **2.3059e-1(5.23e-2)** | 1.1470e+0(7.73e+0) |
|  | 10000 | 1.8414e+5(1.15e+4) | 3.1759e+4(9.53e+3) | 1.7238e+4(1.79e+4) | 2.3949e+3(2.05e+2) | 7.2611e+4(3.76e+2) | 3.0943e-1(4.22e-1) | 3.7318e+0(4.19e+0) | **2.0836e-1(9.05e-11)** | 5.1845e+0(7.19e+0) |
| DTLZ2 | 1000 | 3.5489e+1(5.96e+0) | 4.3729e+0(3.32e-1) | 9.7221e+0(1.67e+0) | 5.4715e+0(7.88e-1) | 1.3163e+0(4.83e-1) | 5.9351e-1(1.21e+0) | **1.0305e-1(4.80e-2)** | 2.1142e-1(2.34e-1) | 3.2014e-1(3.01e-1) |
|  | 5000 | 2.0355e+2(1.08e+1) | 2.5686e-1(2.33e+0) | 4.8258e-1(6.36e+0) | 3.9753e+1(2.40e+0) | 1.3535e+0(3.44e-1) | **6.2989e-2(2.07e-3)** | 3.8928e+0(9.30e+0) | 1.9077e+0(4.45e+0) | 2.4242e+0(5.76e+0) |
|  | 10000 | 4.0705e+2(1.58e+1) | 4.7210e-1(4.39e+0) | 8.5965e-1(1.36e+1) | 7.8984e+1(8.24e+0) | 1.7844e+0(3.13e-1) | **6.1272e-2(3.48e-3)** | 1.8469e-1(2.60e-1) | 8.6939e-2(2.61e-2) | 7.3235e-2(1.65e-2) |
| DTLZ3 | 1000 | 5.8810e+4(5.99e+3) | 9.9866e+3(1.51e+3) | 6.6308e+3(3.05e+3) | 5.7306e+2(4.30e+1) | 2.4370e+4(5.20e+2) | 1.3516e+3(2.39e+3) | 1.3609e+3(2.65e+3) | **2.6458e+2(4.31e+2)** | 2.7038e+2(3.13e+2) |
|  | 5000 | 2.9041e+2(5.79e+3) | 4.3509e+4(1.27e+4) | 3.1891e+4(3.22e+4) | 3.9859e+3(2.17e+2) | 1.2453e+4(5.22e+2) | 2.1861e+3(3.10e+3) | 1.5139e+3(1.68e+3) | **6.1260e-1(2.00e-1)** | 5.0168e+0(5.79e+0) |
|  | 10000 | 5.9058e+5(1.07e+4) | 6.7224e+4(1.33e+4) | 4.9663e+4(4.35e+4) | 8.1286e+3(5.26e+2) | 2.4917e+4(8.08e+2) | **2.7059e-1(4.77e-1)** | 4.0025e-1(9.64e-2) | 5.6419e-1(4.02e-3) | 2.8193e-1(5.35e-1) |
| DTLZ4 | 1000 | 3.6751e+1(6.56e+0) | 8.7989e+0(4.90e+0) | 1.3120e+1(5.38e+0) | 2.7978e+0(1.52e+0) | 1.7943e+0(1.38e+0) | 2.9849e-1(3.52e-1) | 3.0249e-1(2.69e-1) | **1.2255e-1(5.39e-2)** | 3.4578e-1(6.39e-1) |
|  | 5000 | 2.0298e+2(2.29e+1) | 8.0450e+1(2.45e+1) | 7.7825e+1(1.12e+1) | 8.2570e+1(2.04e+1) | 3.1396e+0(2.48e+0) | 3.8238e-1(3.72e-1) | **8.4489e-2(2.91e-2)** | 1.1712e-1(2.69e-2) | 1.3242e-1(4.05e-2) |
|  | 10000 | 3.8442e+2(8.39e+0) | 1.5148e+2(2.89e+1) | 1.3450e+2(1.96e+1) | 1.8845e+2(3.83e+1) | 7.4236e+1(1.51e+2) | 7.2282e-2(1.89e-2) | **6.3001e-2(5.25e-3)** | 1.1355e-1(3.30e-2) | 1.1971e-1(2.15e-2) |
| DTLZ5 | 1000 | 3.5126e+1(6.65e+0) | 3.6667e+0(5.48e-1) | 9.0203e+0(1.74e+0) | 4.8166e+0(4.32e-1) | 9.7212e-1(5.92e-1) | 3.2991e-1(7.55e-1) | 1.5579e-1(3.27e-1) | **2.2791e-2(2.55e-2)** | 5.1728e-1(6.25e-1) |
|  | 5000 | 1.9598e+2(7.66e+0) | 2.6307e+1(2.98e+0) | 4.1106e+1(5.84e+0) | 4.1093e+1(4.26e+0) | 1.7579e+0(7.64e-1) | **6.320e-3(3.44e-3)** | 1.6563e-2(2.83e-2) | 4.9498e-1(1.52e-1) | 3.5783e-1(6.54e+0) |
|  | 10000 | 4.0984e+2(1.33e+1) | 5.1802e+1(5.67e+0) | 8.5696e+1(6.97e+0) | 7.7378e+1(6.35e+0) | 2.0465e+0(9.38e-1) | 5.1561e-3(1.08e-3) | **4.5976e-3(5.98e-4)** | 1.2996e-2(1.00e-2) | 5.2693e-3(7.55e-4) |
| DTLZ6 | 1000 | 6.7254e+2(3.56e+1) | 2.6919e+2(6.47e+1) | 3.2103e+2(1.87e+2) | 4.4008e+2(4.92e+1) | 8.0847e-2(3.6.14e-3) | 8.0427e-3(3.07e-3) | **3.5509e-3(1.61e-4)** | 4.5964e-2(9.10e-3) | 4.5587e-2(1.35e-3) |
|  | 5000 | 3.3458e+3(6.70e+1) | 2.2042e+2(1.67e+2) | 1.7773e+3(6.03e+2) | 1.9879e+3(4.67e+2) | 4.3210e-3(2.55e-3) | 4.3188e-3(3.22e-2) | **3.4626e-3(1.61e-4)** | 2.5415e-2(1.67e-2) | 2.5420e-2(1.94e-2) |
|  | 10000 | 6.8175e+3(2.26e+2) | 4.8254e+3(6.47e+1) | 3.3037e+3(1.63e+3) | 4.6517e+3(2.71e+2) | 8.6914e-3(1.24e-3) | 8.6981e-3(1.55e-3) | **3.4565e-3(6.14e-5)** | 4.9226e-3(3.30e-3) | 4.9357e-3(3.25e-2) |
| DTLZ7 | 1000 | 7.6884e+0(1.14e+0) | 9.5440e+0(3.35e-1) | 9.9196e+0(2.93e-1) | 7.8822e+0(5.09e-1) | **1.1498e-1(1.15e-1)** | 2.0752e-1(8.20e-2) | 2.0377e-1(5.91e-2) | 6.5215e-1(1.70e-1) | 6.6534e-1(1.93e-1) |
|  | 5000 | 9.6590e+0(8.82e-1) | 1.1118e+1(7.16e-1) | 1.0916e+1(3.29e-2) | 1.0017e+1(3.73e-1) | **2.5499e-1(1.44e-1)** | 6.2057e-1(6.31e-2) | 6.2603e-1(9.97e-2) | 1.1193e+0(1.59e-1) | 1.1121e+0(9.66e-2) |
|  | 10000 | 1.0748e+1(3.13e-2) | 1.1279e+1(1.04e-1) | 1.1170e+1(8.66e-2) | 1.0490e+1(1.73e-1) | **4.0907e-1(3.34e-1)** | 6.0830e-1(3.39e-2) | 8.0574e-1(5.30e-2) | 1.0420e+0(1.14e-1) | 1.0410e+0(8.64e-2) |

