# OpenReview forum: "Learning Deep Improvement Representation to Accelerate Evolutionary Optimization"
_ICLR.cc/2024/Conference — ICLR 2024 Conference Withdrawn Submission_

### Official Review · Reviewer_u1rY · 2023-10-29

**Soundness:** 2 fair
**Presentation:** 2 fair
**Contribution:** 2 fair
**Rating:** 3
**Confidence:** 4

**Summary:**

This paper introduced a machine learning enhanced evolutionary algorithm for solving multi-objective optimization problems. The key idea is to obtain a high-level and more compressed representation of the solution space such that evolutionary search can be effectively carried out in the learned feature space, instead of in the original large solution space. This is expected to not only improve the algorithm performance but also its efficiency. The usefulness of this idea has been experimentally evaluated on several commonly used benchmark multi-objective optimization problems.

**Strengths:**

Various machine learning techniques have been increasingly utilized to improve the performance of evolutionary algorithms in recent years. This paper introduced another possible way of using neural network models to learn high-level solution representations that will guide evolutionary search towards identifying Pareto optimal solutions.

**Weaknesses:**

While the idea of conducting evolutionary search in the learned representation space, rather than the original solution space, is interesting, there is no guarantee at the theoretical level why search in the representation space is likely to be more efficient and effective. Empirical studies based on a few benchmark problems alone are not sufficient to demonstrate the wide applicability of this idea and its theoretical validity. In addition, if the goal is to train the neural network to produce a high-level representation that facilitates evolutionary search, should this be considered as part of the loss function for training the neural network? Why or why not? These questions must be answered explicitly to clearly understand the novelty of using representation learning techniques in evolutionary algorithms.

One aim of this paper is to enhance the efficiency of evolutionary algorithms. However, I was wondering how efficiency is actually defined and measured. If the idea of efficiency is mainly about computation time or computation complexity, training even a shallow neural network may not necessarily be computationally efficient. Note that the hidden layer might be small, however the input layer and the output layer are at the same scale as the dimensionality of the solution space, which can be quite high for real-world problems. Hence, I am not fully convinced by the statement on page 5 that "the computational complexity of running this model is akin to traditional evolutionary search operators", especially when multiple shallow networks are stacked together to build a deep neural network to produce high-level solution representations. The actual computation complexity of the new algorithm, compared to existing evolutionary algorithms, must be analyzed mathematically.

In line with the above concern, it appears to me that the authors believe stacking multiple trained shallow networks together can help to extract useful high-level solution representations. I doubt to which extent this idea may work and why. Theoretically, I cannot find direct clues in this paper regarding why stacking multiple trained shallow networks can produce useful high-level representations. There is a lack of theoretical guarantee for this stacking trick. On the other hand, if the full network must be trained end-to-end after stacking, the computation complexity may be substantially increased, affecting the usefulness of the new algorithm.

The correctness of some formulas should be checked further. The clarity of some formulas should also be improved. For example, I don't understand eq. (2). Particularly, how can the argmin operator over D produce a tuple that not only contains D but also A and $\theta$?

It also appears confusing to me why this paper focuses on multi-objective optimization problems. Upon considering single-objective optimization problems, the importance of learning high-level solution representations should not be overlooked. Compared to multi-objective optimization problems, it might be easier to study the usefulness of learning high-level solution representations for solving single-objective optimization problems. In view of this, the primary research question that this paper aims to address should be clearly presented and justified.

Some critical parameter settings such as the learning rate should be clearly justified. The proposed learning rate of 0.1 is a bit larger than the learning rate commonly used to train neural networks in the literature. Other parameter settings related to the momentum and number of epochs should be clearly justified too.

Some statements are hard to understand. For example, for the statement on page 6 "completely avoiding search ... leading to slow growth of the MLP model", what does it mean? How will the MLP model grow while running the new algorithm? Why will the growth of the MLP depend on search in the original solution space?

The experiments only considered benchmark problems with two and three objectives. Why didn't the authors consider problems with more than three objectives? Multi- and many-objective optimization is a hot topic for evolutionary computation. Many new algorithms have been developed in the past few years to solve similar benchmark problems studied in this paper. The authors are suggested to include several more recently published baselines in the experimental study.

**Questions:**

Theoretically, why can the evolutionary search in the representation space be more efficient and effective?

If the goal is to train the neural network to produce a high-level representation that facilitates evolutionary search, should this be considered as part of the loss function for training the neural network? Why or why not?

When multiple shallow networks are stacked together to build a deep neural network to produce high-level solution representations, why is the computational complexity of running this neural network akin to traditional evolutionary search operators?

Theoretically, why is it possible to produce useful high-level representations by stacking multiple trained shallow networks?

Why should this paper focus on multi-objective optimization problems rather than single-objective problems?

---

### Official Review · Reviewer_esui · 2023-10-30

**Soundness:** 2 fair
**Presentation:** 2 fair
**Contribution:** 1 poor
**Rating:** 3
**Confidence:** 5

**Summary:**

This proposes a lightweight model for learning improvement representations to enhance evolutionary algorithms for LMOPs. A stacked autoencoder is used for this purpose. The proposed method is tested against two classic multi-objective optimization methods NSGA-II and MOEA/D on several benchmark functions.

**Strengths:**

The results on the tested benchmarks are good. Solving 10000-d problems with 10^5 evaluations look promising.

**Weaknesses:**

The proposed stacked MLPs seem a straigtforward application of the MLP proposed by innovization progress learning (Mittal et al., 2021a). What does the stacked MLPs really provide to the search seem quite vague. For example, it is said "and may not fully exploit the potential of evolutionary search". What does this mean by potential? This makes the contribution of this work not very clear.

**Questions:**

1. From the results, especially the results of v2-v5, the proposed method  enjoys very much faster convergence rate at the begining of the search, indicating the stacked MLP contribute a lot. Why the probability of choosing the MLP is reducing as the search goes on as shown in alg.3?
2. What does the stacked MLP exactly learn? Detailed analysis and explaination should be given.
3. It is not sure how significant are the results on the benchmark. Why do we test on those benchmarks? What characteristics do they representation of the real-world problems?
4. From Table 2, it is unsure how significant the improvements are on the HV indicator.
5. What is the latent space to be searched within the stacked MLP? There seems to be lots of latent space.

---

### Official Review · Reviewer_APTU · 2023-11-03

**Soundness:** 1 poor
**Presentation:** 2 fair
**Contribution:** 2 fair
**Rating:** 5
**Confidence:** 4

**Summary:**

This paper proposes a novel use of the deep neural networks into multi-objective evolutionary algorithms (MOEAs). The overall search mechanism is the same as standard MOEAs, except that candidate solutions are modified through the auto-encoder-like network model, aiming at improving them in terms of Pareto optimality. In this sense, the proposed algorithm works as a memetic algorithm: a combination of evolutionary computation and local search, the latter of which is used to improve candidate solutions locally. The proposed algorithm variants are constructed on top of existing NSGA-II and MOEA/D and they has been compared with the baseline approaches, NSGA-II and MOEA/D, on DTLZ testbed and the time-varying ratio error estimation (TREE) problems.

**Strengths:**

A novel use of DNNs in MOEAs.

DNN architectures and the loss for the training of DNN has been proposed.

Promising experimental performance has been shown.

**Weaknesses:**

Test Problems. DTLZ test problems are very notorious with their unrealistic feature in evolutionary computation communities. The variables are decomposed into two groups: ones determining the distance to the Pareto front, and ones determining the spread into the Pareto front. See for example (*). It is very unrealistic and good results on DTLZ do not usually imply good performance on other problems as an algorithm can easily overfit. Because of this feature, I am very suspicious of validity of evaluating the proposed approach on DTLZ problems. In particular, I have a feeling that large scale (high dimensional) DTLZ do not reflect the difficulties of other large scale multi-objective optimization problems as most variables only needs to converge to some point to obtain good performance. The authors should consider to add more experimental evaluation on other testbeds such as WFG.

(*) https://doi.org/10.52731/ijscai.v2.i1.285

The proposed approach is a kind of memetic algorithm. However, the comparison is only done without any repair mechanism. To show the goodness of the proposed approach, the authors should consider comparing the proposed approach with memetic algorithms with the same baseline MOEAs.

The proposed approach has been applied to TREE problems, showing promising results. However, because the problem characteristics is not described at all in the paper, the reason of the good performance of the proposed approach over the baseline approach is not understandable. Please provide the problem characteristics and the reason why the authors expect that the proposed approach works better than the baselines in relation to the problem characteristics. Otherwise, this result does not support the goodness of the proposed approach.

**Questions:**

Please answer the above comments.